# Brain Functional Representation of Highly Occluded Object Recognition

**DOI:** 10.3390/brainsci13101387

**Published:** 2023-09-29

**Authors:** Bao Li, Chi Zhang, Long Cao, Panpan Chen, Tianyuan Liu, Hui Gao, Linyuan Wang, Bin Yan, Li Tong

**Affiliations:** Henan Key Laboratory of Imaging and Intelligent Processing, PLA Strategic Support Force Information Engineering University, Zhengzhou 450001, China; libao0108@outlook.com (B.L.); zcboluo@hotmail.com (C.Z.); tytauceti@hotmail.com (T.L.);

**Keywords:** fMRI, MVPA, PPI, dACC, occipital lobe, occluded object recognition

## Abstract

Recognizing highly occluded objects is believed to arise from the interaction between the brain’s vision and cognition-controlling areas, although supporting neuroimaging data are currently limited. To explore the neural mechanism during this activity, we conducted an occlusion object recognition experiment using functional magnetic resonance imaging (fMRI). During magnet resonance examinations, 66 subjects engaged in object recognition tasks with three different occlusion degrees. Generalized linear model (GLM) analysis showed that the activation degree of the occipital lobe (inferior occipital gyrus, middle occipital gyrus, and occipital fusiform gyrus) and dorsal anterior cingulate cortex (dACC) was related to the occlusion degree of the objects. Multivariate pattern analysis (MVPA) further unearthed a considerable surge in classification precision when dACC activation was incorporated as a feature. This suggested the combined role of dACC and the occipital lobe in occluded object recognition tasks. Moreover, psychophysiological interaction (PPI) analysis disclosed that functional connectivity (FC) between the dACC and the occipital lobe was enhanced with increased occlusion, highlighting the necessity of FC between these two brain regions in effectively identifying exceedingly occluded objects. In conclusion, these findings contribute to understanding the neural mechanisms of highly occluded object recognition, augmenting our appreciation of how the brain manages incomplete visual data.

## 1. Introduction

In recent years, with the emergence of deep learning, computer vision algorithms have demonstrated the ability to handle simple object recognition tasks [1]. However, accurate recognition faces significant challenges when the objects are highly occluded. These challenges primarily stem from information loss, where critical contextual information is missing due to occlusion, making it difficult for algorithms to accurately detect and recognize objects [2]. However, humans have unique cognitive abilities in highly occluded object recognition [3]. Achieving human-level performance in object recognition tasks serves as an important milestone for computer vision algorithms. By reaching or surpassing human performance levels, we can assess the robustness and adaptability of algorithms in handling complex real-world scenarios [4,5]. Research on the neural mechanisms of humans during object recognition under high occlusion holds the potential to provide valuable guidance for computer vision algorithms in addressing similar problems [6].

Recently, an anatomical model for occluded object recognition was proposed via the Golin incomplete image recognition task [7], dividing object recognition into two primary stages: visual processing and semantic classification. This model provides a basic characterization of the two stages involved in recognizing occluded objects but does not delve into a detailed analysis at the level of brain voxels. According to one functional magnetic resonance imaging (fMRI) study, low-level visual regions exhibited more activation in the occluded condition as compared to the non-occluded one [8]. Another study suggested that low-level visual regions encoded only spatial information about occluded objects, while high-level visual regions were responsible for representing their identities [9]. Although these fMRI studies provided detailed insight on object processing under occlusion, they mainly study information transfer and processing in visual areas of the brain without mentioning other brain areas [10].

In the context of highly occluded object recognition, the involvement of the prefrontal cortex (PFC) has been shown to be crucial. The PFC is a brain region responsible for higher-order cognitive processes such as working memory, decision-making, and goal-directed [11]. Studies utilizing event-related potentials (ERPs) have demonstrated the PFC’s involvement in integrating visual features of occluded objects [12]. Furthermore, the PFC is involved in allocating attentional resources to occluded objects, improving recognition by maintaining relevant representations in working memory [13]. Neurophysiological recordings in primates have shown that signals between the PFC and visual cortex interact to recognize objects under occlusion [14]. Anatomical research has also revealed direct projections from the visual cortex to the PFC and reciprocal connections [15,16]. While previous studies have identified the involvement of the PFC and occipital regions in object recognition under occlusion, further research is needed to explore the extent of their involvement under different occlusion conditions and the interactions between these brain regions.

The excellent spatial resolution of fMRI can assist us in studying subtle changes in specific brain regions during the task state [17]. Generalized linear model (GLM) analysis is a flexible and powerful method for analyzing complex brain signal data, enabling the identification of specific brain activation regions related to a given task [18]. For the specific brain regions resulting from the GLM analysis, their statistical significance differences can be further investigated using the multivariate pattern analysis (MVPA) [19]. Psychophysiological interaction (PPI) analysis is to further investigate the functional connectivity between these specific brain regions [20]. Figure 1 presents the roadmap of the whole study.

The focus of our study was to gain a deeper understanding of the neural mechanisms involved in recognizing highly occluded objects. To accomplish this objective, we introduced three-degree occluded image recognition tasks within an fMRI environment. Our research goes beyond previous work by not only investigating activation patterns in the brain but also exploring the functional connectivity between different brain regions during different occluded object recognition. Utilizing techniques such as GLM, MVPA, and PPI, we were able to elucidate the functional representation of highly occluded object recognition in the brain. These analytical methods facilitated the exploration of three fundamental questions: (1) What is the role of the prefrontal cortex (PFC) and occipital regions in processing highly occluded objects? (2) Which specific brain areas are involved in this process? (3) Are there notable differences in the interactions and contributions of these regions under varying degrees of occlusion?

## 2. Methods

### 2.1. Participants

Altogether, 66 university students participated in this study, and two subjects were excluded from the final fMRI analysis due to poor imaging quality. Eventually, 64 subjects (32 women, mean age of 22 years, aged between 19 and 26 years) were included in the final analysis. All participants were right-handed, and screening using the anxiety self-rating scale indicated no presence of any psychiatric disorders. Additionally, none of the participants had used any medication in the month prior to the experiment. All subjects completed an informed consent form and an MRI safety screening questionnaire before entering the MRI scanner and received compensation for their participation. The Ethics Committee of Henan Provincial People’s Hospital granted approval for all experimental design protocols. All research procedures were in agreement with related standards and regulations.

### 2.2. Stimuli

The initial experimental stimulus material was sourced from the publicly accessible SynISAR simulation dataset [21], from which two aircraft images were chosen as recognition objects. We utilized the Pixlr software(Pixlr X) (https://pixlr.com/cn/URL (accessed on 1 January 2022) to make adjustments to the target images. The pixel dimensions of each image were standardized to 678 × 535, and the contrast enhancement effect was set to zero to ensure similarity among all stimulus pictures. In order to produce an occluded image, black squares of varying sizes were iteratively added at random positions on the image. Taking reference from a recent study on occluded object recognition [6], we defined three levels of occlusion: 10% (baseline conditions), 70%, and 90% occlusion. Instead of using a completely clear target, we used the 10% occlusion as the baseline condition to eliminate any potential interference from the black squares. The resulting image set consisted of six conditions: 2 targets × 3 levels of occlusion (Figure 2A). For each condition, fifty sample images were generated by altering the orientation of the aircraft and the position of occlusion.

### 2.3. fMRI Experimental Paradigm

Before the MRI scan, we provided participants with instructions about the task, and they practiced the task on an external computer with the exact same user interface as the MRI scanning experiment. However, the practice task may allow the participants to memorize the target in advance, affecting the effectiveness of the experiment. To avoid this situation, the target images used in the practice task were different from those used in the MRI scanning experiment.

The experimental paradigm (Figure 2B) was designed in reference to the Gollin incomplete image recognition task [7]. As the subjects were in the scanner, the process began with anatomical scans, succeeded by functional scans, totaling approximately 50 min for each subject. The participants were asked to view and memorize the features of two clear aircraft targets prior to each of the two experimental runs. Each run comprised 30 blocks, within which each task (10%, 70%, and 90% occlusion) presented 10 blocks randomly. Each block maintained a consistent occlusion level and contained five trials, with either aircraft A or aircraft B appearing at random. The stimulus presentation strategy used in this study was a pseudo-randomization strategy. The three tasks were shuffled in a predetermined order to achieve a random presentation of the tasks while maintaining consistency in the sequence seen by each participant.

Before each run, the subjects were asked to watch two clear aircraft objects to remember their features. Each run consisted of 30 blocks, of which 10 blocks were randomly presented for each of the three tasks (10%, 70%, and 90% occlusion). The target image occlusion level of each block was the same, which contained five trails, with either aircraft A or aircraft B randomly appearing in each trail. The experiment required a simple binary decision from the subjects, who had to discern whether the viewed image was aircraft A or B. Responses were recorded via button press: the right index finger for aircraft A and the right middle finger for aircraft B (Figure 2C).

### 2.4. MRI Data Acquisition

Functional and structural MRI scans were conducted using a standard 64-channel head coil on a 3T Siemens Magneto Prisma scanner (manufactured by Siemens Healthineers, German). Subjects observed the projection screen using a mirror system mounted on the head coil. Functional images were captured using a single-shot gradient echo and echo-planar imaging (EPI) sequence using the parameters: TR of 2000 ms, TE of 30 ms, flip angle of 76°, a 96 × 96 matrix, field of view (FOV) of 192 × 192 mm^2^, and 2 × 2 × 2 mm^3^ voxel size resolution. Each EPI volume included 64 transversal slices (2 mm thick with no gap) in an interleaved order containing 504 functional images per run. For localization of activation assistance, we obtained high-resolution T1-weighted magnetization-prepared rapid gradient-echo imaging (MP-RAGE) 3-D MRI using a pulse sequence with TR of 2300 ms, TE of 2.26 ms, flip angle of 8°, 256 × 256 matrix, FOV of 256 × 256 mm^2^, 1 × 1 × 1 mm^3^ voxel size resolution, and 192 contiguous axial images each 1 mm thick.

### 2.5. MRI Data Preprocessing

We employed Statistical Parametric Mapping software (SPM12; Wellcome Department of Cognitive Neurology, London, UK) to preprocess and conduct statistical analysis on the functional images. During preprocessing, the EPI data were corrected for slice time and head movement relative to the median functional volume. Subsequently, the data were co-registered and normalized to align with the standard Montreal Neurological Institute (MNI, McGill University, Montreal, QC, Canada) coordinate space. Lastly, we applied a Gaussian kernel for spatial smoothing, with a full width at half maximum (FWHM) specification of 6 mm.

### 2.6. GLM Data Analysis

The conducted GLM data analysis utilized SPM12 to reveal regions activated by the occlusion stimulus, which involved two different methods of evaluation: Firstly, the analysis models integrated all three tasks, and secondly, each task was modeled separately to discern the effects across the tasks. Contrast images for the main effects of three tasks were generated for each participant and subjected to a second-level analysis for population-level inference. A threshold for the statistical results was defined at *p* < 0.05, as corrected using FWE.

To visualize and further analyze the different levels of occlusion-dependent response, a region of interest (ROI) analysis was performed in this study. Referring to the GLM results acquired afore, occipital lobes and the dACC were chosen as the ROIs, and the Marsbar 0.45 toolbox was utilized to extract the mean Betas from them. The derived betas were then subjected to repeated-measures *t*-tests with FWE correction to underscore significantly different regional responses to varying occlusion levels.

### 2.7. Multivariate Pattern Analysis

The classification of three occluded object recognition tasks was carried out across subjects, utilizing the Python 3.90-based PyMVPA toolbox [22]. A reiteration of the GLM analysis was performed for each subject to generate spmT files for each block, resulting in an aggregate of 3840 samples (64 subjects × 60 blocks). Referring to the GLM results acquired afore, brain regions of bilateral occipital lobes (inferior occipital gyrus—IOG, middle occipital gyrus—MOG, and occipital fusiform gyrus—OFG) and dACC in AAL90 were selected as templates to segment the brain regions of the spmT file, and the mean beta values of each brain region were extracted to generate a 7 × 1 feature vector. The beta values underwent normalization (µ = 0, σ = 1) prior to the MVPA application. We trained a Support Vector Machine (SVM) classifier, using a Radial Basis Function (RBF) kernel, to categorize three separate occluded object recognition tasks using leave-one-out cross-validation, where the model was trained on the data from all except one subject and tested on the hold-out subject data. We determined the classifier’s accuracy by computing the proportion of accurately classified events relative to the overall number of events.

To compare the functions of the dACC and the occipital lobe, ablation experiments were carried out where dACC features were excluded. It is worth noting that, in order to eliminate the impact of feature quantity on decoding accuracy, we did not simply remove the features of dACC in the ablation experiment. Instead, we used the random permutation method to estimate the null distribution. In this approach, we randomly shuffled the order of dACC features 1000 times in the ablation experiment. This approach ensures that the ablation experiment and the original classification experiment have the same number of features while eliminating the enhancement effect of dACC features.

To verify the significance of the difference in classification accuracy between the ablation experiment and the original experiment, for the accuracy of the classification experiments with different features, we performed a two-sample *t*-test with FDR correction.

### 2.8. gPPI Data Analysis

Functional connectivity (FC) refers to the meaningful interactions and communication between different brain regions, revealing the coordinated workings of the brain’s structure and function. To determine the correlation between the FC of the dACC and the occipital lobe with the occlusion level, we employed the generalized form of the PPI (gPPI) method [23,24], an approach ideally suited to scrutinizing FC during block-design experiments. Traditional PPI analysis is able to calculate functional connectivity for specific ROI between all voxels in the whole brain, while gPPI can calculate functional connectivity between two ROIs in different blocks. Based on whole-brain activation analysis findings, we selected bilateral occipital lobes (IOG, MOG, and OFG) and dACC as ROIs. For each participant, the ROIs’ time series were extracted using the first eigenvariate of a 6 mm radius sphere surrounding the peak (Table 1). At the individual participant level, the inclusion criteria were: (1) a psychological variable signifying the three occlusion levels; (2) a physiological variable of the ROIs; and (3) PPI values, the cross product of the first two regressors. To obtain FC values between the dACC and occipital lobes, we computed regression coefficients for the PPI values of the two relevant ROIs. Subsequently, we averaged the FC values between the dACC and six occipital lobes for each participant across all three tasks. This was followed by conducting a paired *t*-test on the averaged FC values between the dACC and occipital lobes. Because the PPI value benefits from the physiological variable (the first BOLD eigenvector), to demonstrate the reliability of the PPI value, we further calculated the Pearson correlation coefficient for the first BOLD eigenvector between the dACC and other occipital regions in different occlusion levels. We tested the significance of the difference between each of the two occlusion levels using a two-sample *t*-test under the FDR correction condition.

## 3. Results

### 3.1. Behavioral Data

For the object recognition tasks, the average accuracy at occlusion levels of 10%, 70%, and 90% stood at 96%, 79%, and 62%, respectively (Figure 3A). The average reaction times were 1.23 s, 1.65 s, and 1.89 s, respectively (Figure 3B). We tested the significance of the difference between each of the two occlusion levels using a two-sample *t*-test under the FDR correction condition. These figures indicated that there were significant disparities in participants’ performance both regarding the accuracy and reaction time (FDR, two-sample *t*-test: *p* < 0.001), which amplified as the degree of occlusion rose.

### 3.2. GLM Analysis Results

Comparative brain activation maps between the occluded object recognition task and the resting state demonstrated discernible activation in the dACC, along with the occipital, parietal, and frontal lobes during the tasks (Figure 4A). Tasks with high levels of occlusion (70% and 90%) corresponded with stronger activation in the dACC and occipital lobe (IOG, MOG, and OFG) relative to tasks with low levels of occlusion (10%) (Figure 4B,C). Notably, dACC activation continued to strengthen as the occlusion level increased from 70% to 90% (Figure 4D). Table 1 provides the MNI coordinates of the differentially activated brain regions.

To achieve a clearer representation and deeper analysis of brain responses to varying tasks, we performed quantitative comparisons of these ROIs. We used a two-sample *t*-test under the FWE correction condition to analyze the significance of the differences. Figure 5 delineated beta values of brain regions that showed notable main effects during the GLM analysis. Within the occipital lobe, tasks involving 70% and 90% occlusion induced stronger activation (Figure 5A–C). Notably, only tasks with 90% occlusion triggered intensified activation in the dACC (Figure 5D).

### 3.3. Multivariate Pattern Analysis Results

The presence of brain activation variations in the dACC and the occipital lobe was observed across all three tasks, which led us to further investigate their potential to elicit statistically significant activation patterns. We first employed features from bilateral occipital lobes and the dACC for classification, resulting in an average accuracy of 53.7% (Figure 6A). We then conducted an ablation experiment by randomly shuffling the dACC features 1000 times, which led to a decrease in the average classification accuracy to 46% (Figure 6B). The two-sample *t*-test in the FDR correction condition demonstrated that shuffling the dACC features significantly decreased classification accuracy at 70% and 90% occlusion levels (Figure 6C). To validate the performance of both models, we calculated the F1-score value, which dropped from 0.87 to 0.83 after shuffling the dACC feature, further demonstrating the importance of the dACC feature for the model classification performance.

### 3.4. Functional Connectivity: gPPI Analysis Results

To investigate the interplay between the dACC and occipital lobe, we calculated the FC values between them using the gPPI method. The slope of the straight line indicated the PPI values between the dACC and left IOG for subject 1 in the three tasks (Figure 7C). Similarly, we calculated the PPI values between the dACC and the other six occipital ROIs (Figure 7B) for each subject in three tasks. Group-level analysis revealed that the PPI values between dACC and occipital lobe increased with increasing occlusion level (Figure 7D), and these differences were determined to be statistically significant.

The first BOLD eigenvector results further demonstrated the reliability of the PPI results. By comparing the pearson correlation coefficients of the time series between the dACC and other brain regions at different occlusion levels, we observed that the pearson correlation coefficient was the lowest under the 10% occlusion level, while it exhibited a significant increase and statistical significance under the 70% and 90% occlusion levels (FDR, two-sample *t*-test: * = *p* < 0.05, ** = 0.01, *** = 0.001) (Figure 8). These findings are similar to the results from the PPI analysis (Figure 7D), further supporting the correlation between occlusion levels and functional connectivity in the brain regions.

## 4. Discussion

In order to assess the role of the PFC in the recognition of partially occluded objects, we studied the functional representation of the dACC and occipital lobe when humans recognized objects at different occlusion levels. By employing behavioral data, GLM analysis, MVPA, and PPI analysis, we were able to underline the essential function of the dACC and occipital lobe in recognizing highly occluded objects.

The unique advantage of our study, in contrast to previous research, was the investigation of the involvement of the PFC and occipital regions in recognizing occluded objects. In comparison, previous studies were limited to the role of visual areas in occluded object recognition. Furthermore, we provided additional evidence using MVPA that the dACC plays a crucial role in successfully recognizing highly occluded objects. By comparing the baseline tasks (10% occlusion), we found that higher occluded conditions (70% and 90% occlusion) were related to stronger activation and FC in the dACC and occipital lobe, which provided evidence for the significance of these brain regions in processing incomplete information.

### 4.1. The Activation of the dACC and Occipital Lobe under Different Occluded Conditions

Our findings indicated that stimuli with higher levels of occlusion (70% and 90%) elicited stronger activation of the occipital lobe compared to the baseline tasks with only 10% occlusion (Figure 4B,C and Figure 5A–C), which were consistent with some reports that studied the effect of occlusion degree on visual area [25,26]. In highly occluded situations, the reduction of available visual information mainly interfered with the down-top processes so heightened activation of the visual area was required to recognize occluded objects [27].

However, instead of eliciting a heightened activation in the occipital lobe compared to the 70% occlusion, the 90% occlusion elicited stronger activation in dACC (Figure 4D), which may be explained using the S-type activation theory in neural cytology [28]. The S-Type Activation Theory proposes that certain brain regions, such as the occipital lobe, have a threshold for processing specific information. In the case of visual information processing, the occipital lobe reaches its threshold at the occlusion level of 70%. Beyond this threshold, the demands placed on visual processing exceed the capacity of the occipital lobe alone. We believed that the occipital lobe reached the threshold of visual information processing at the occlusion level of 70%, necessitating the dACC’s heightened activation to meet the needs of a 90% occlusion task. This inference was also confirmed using MVPA results (Figure 6), which showed that features of the dACC played a significant role in classifying the 90% occluded task from the other tasks.

### 4.2. The FC between the dACC and Occipital Lobe

Previous research on task-state FC demonstrated an overall increase in across-network FC during cognitive tasks [29]. It was also observed that enhancement of across-network FC in specific brain regions was related to task difficulty [30]. The dACC and occipital lobe, belonging to distinct brain networks, synchronously activated to perform highly occluded tasks (Figure 4), and these brain regions may constitute the key network for occluded object recognition. To reveal their interaction, a PPI analysis was executed, leading to the discovery that the FC between the dACC and occipital lobe consistently intensified as the degree of occlusion progressed from 10% to 90% (Figure 7D). As cited in earlier research, the human brain transitions into a state of superior global integration to resolve more complex tasks [31]. Analogously, the amplified FC between the dACC and occipital lobe promoted information interchange, facilitating the processing of high occlusion tasks.

### 4.3. The Role of the dACC in Decision-Making Processes under Highly Occlusion

To perform the occluded object recognition task in the study, the test stimulus on the screen must be compared to the reference stimulus held in memory. Given the function of PFC in working memory, it serves as a reasonable neural site for facilitating such comparison [32]. The results suggested that the heightened activation and FC of the dACC for highly occluded objects might be aligned with the dACC’s decision-making role [33]. Subjects in this experimental paradigm were required to scrutinize an occluded image that appeared on the screen and subsequently make a decision by pressing a button. In the highly occluded tasks, subjects took more time (Figure 3B) for decision-making and obtained lower accuracy (Figure 3A). Decision-making becomes complicated when the visual cortex receives less information as a result of occlusion [34]. In this case, the dACC might amplify weak signals to facilitate decision-making by increasing its own activation and the FC with visual areas. In this regard, our results were consistent with the dACC engaging in decision-making tasks under uncertainty [35,36]. Therefore, the decision-making role of dACC under uncertain conditions might be the key reason why humans can efficiently identify highly occluded targets.

## 5. Conclusions

In summary, our research suggests that the dACC in the PFC and certain occipital lobes, including the IOG, MOG, and OFG, play a crucial role in the recognition process of highly occluded objects. We found that under higher occlusion conditions, there was greater activation and interaction observed in the dACC and occipital lobe. While the occipital regions are responsible for the initial extraction and processing of visual features, the PFC may assist in decision-making when the task becomes more challenging. The role of the dACC in handling increased demands during tasks with higher occlusion may provide insights into the importance of decision-making processes in computer vision algorithms.

## 6. Limitation

In this study, we used a sample size of 66 participants, which is a reasonable sample size for many neuroimaging studies. However, in terms of generalizability, this sample size may still be limited. Larger and more diverse samples could strengthen the conclusions of the study. The study focused on object recognition tasks with varying degrees of occlusion. While this is informative for understanding the neural mechanisms behind this specific task, it may not be applicable to other cognitive processes or real-world situations. To enhance the ecological validity of the findings, future research could employ a combination of fMRI and virtual reality devices. Although this study identified correlations between brain regions and the degree of occlusion, it does not establish causation. Further research using intervention measures would be needed to determine causality. fMRI has limited temporal resolution, which means it cannot capture rapid brain processes. To better understand the temporal dynamics of object recognition, future studies could integrate EEG and MEG methods.

## Figures and Tables

**Figure 1 brainsci-13-01387-f001:**
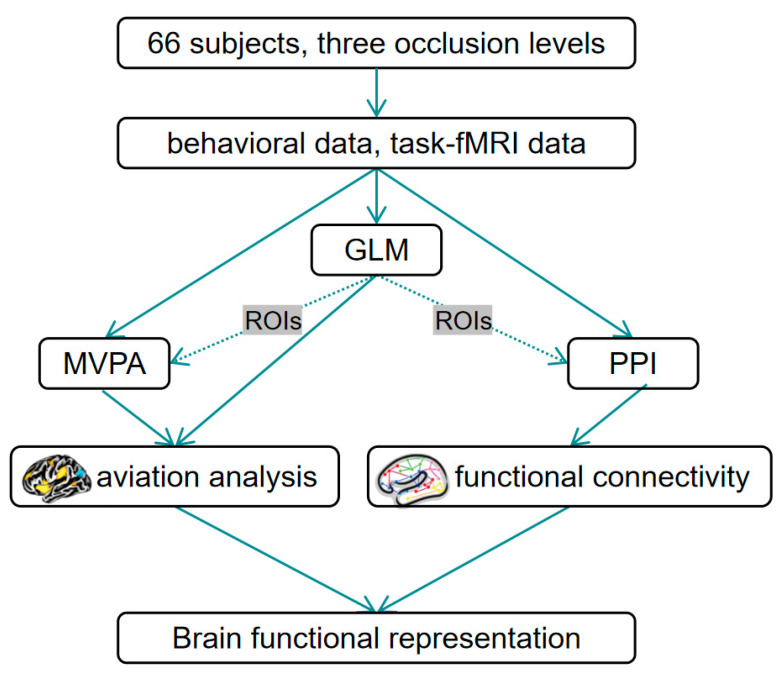
The roadmap of the study.

**Figure 2 brainsci-13-01387-f002:**
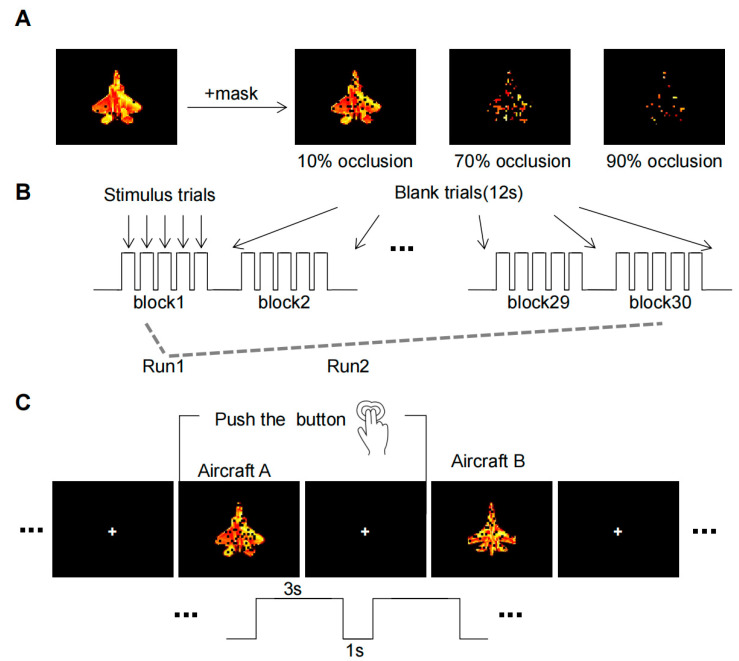
The fMRI experimental paradigm. (**A**) Occlusion Image Set. Aircraft targets undergo three levels of occlusion to generate an image set. (**B**) The whole process of MRI scans. Three types of occlusion image stimuli (10%, 70%, and 90%) were randomly presented in 10 blocks in each run. (**C**) Task operations in each trail. Subjects must make a decision within a 4-s window when the object is displayed and indicate their choice by pressing a button.

**Figure 3 brainsci-13-01387-f003:**
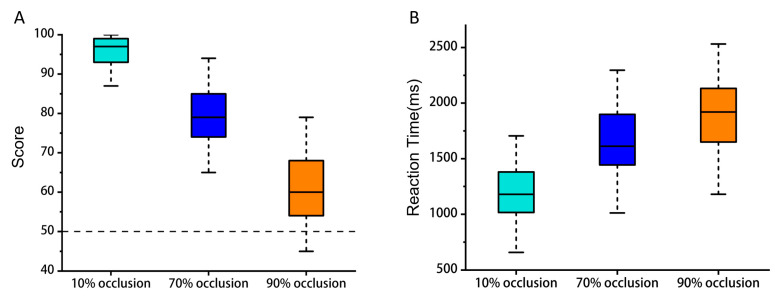
Behavioral results. (**A**) Mean accuracy for object recognition tasks over three occlusion degrees. (**B**) Mean reaction time for object recognition tasks over three occlusion degrees. (FDR, two-sample *t*-test: *p* < 0.001).

**Figure 4 brainsci-13-01387-f004:**
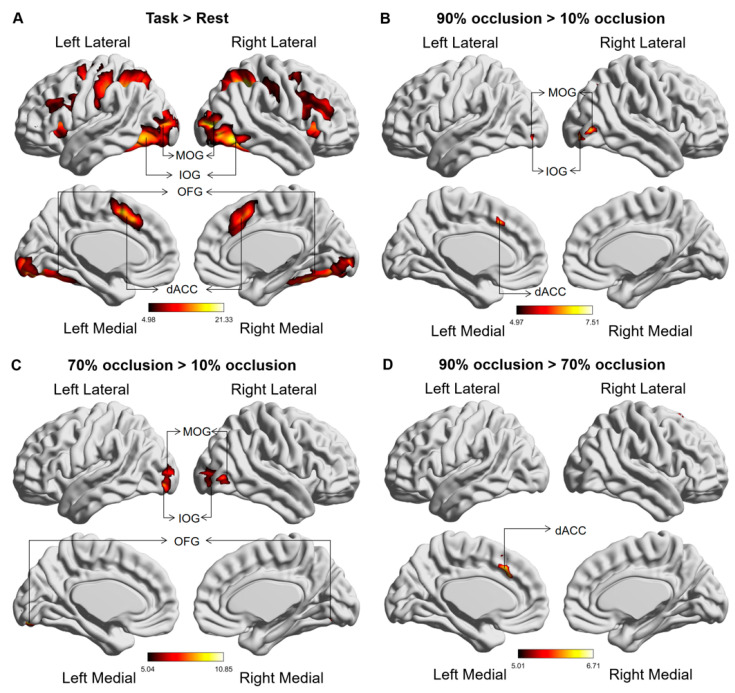
Global brain activation of the group analysis. (**A**) Brain activation maps in comparison between the occluded object recognition task and the resting state. (**B**) The variation in activated brain regions between tasks at 90% occlusion and 10% occlusion. (**C**) The variation in activated brain regions between tasks at 70% occlusion and 10% occlusion. (**D**) The variation in activated brain regions between tasks at 90% occlusion and 70% occlusion. The color bar represents the t value. The abbreviations used are as follows: IOG—Inferior Occipital Gyrus, MOG—Middle Occipital Gyrus, OFG—Occipital Fusiform gyrus, dACC—Dorsal anterior cingulate cortex.

**Figure 5 brainsci-13-01387-f005:**
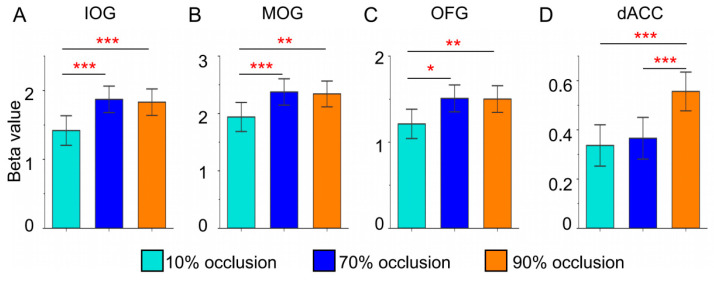
Brain regional (IOG, MOG, OFG, and dACC) responses (betas) to three types of occluded objects. (FWE, two sample *t*-test: * = *p* < 0.05, ** = *p* < 0.01, *** = *p* < 0.001).

**Figure 6 brainsci-13-01387-f006:**
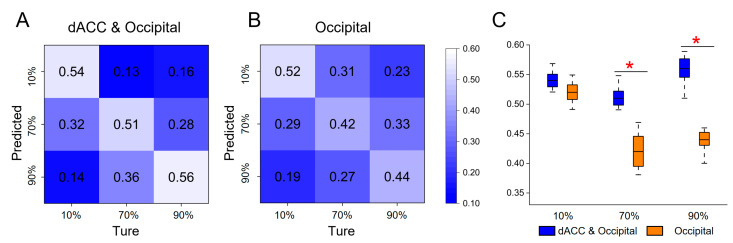
The results of multivariate pattern analysis. (**A**) Confusion matrix for classifying three types of occlusion image stimuli (taking both occipital and dACC brain activation as features). (**B**) Confusion matrix for the ablation experiment (randomly shuffling the dACC features). (**C**) Differences in accuracy between the above two classification experiments (baseline = 33%). (FDR, two-sample *t*-test: * = *p* < 0.05).

**Figure 7 brainsci-13-01387-f007:**
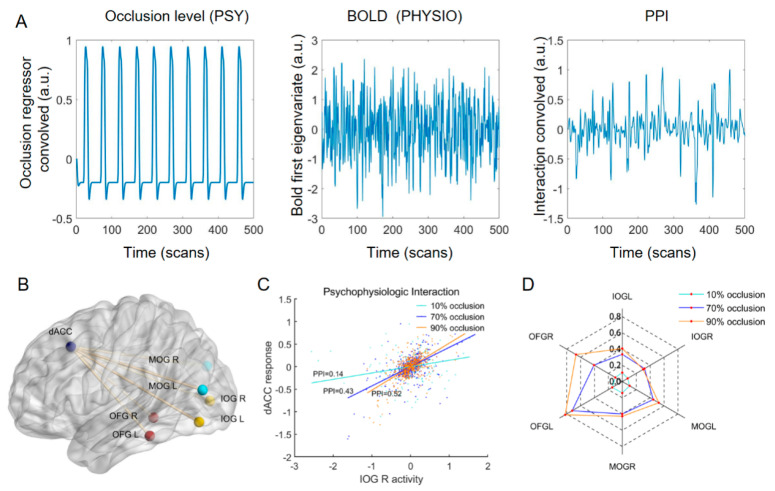
PPI results. (**A**) Representative examples of the regressors used for the PPI analyses include the psychological regressor (occlusion level; left subplot), physiological regressor (BOLD first eigenvariate of the ROIs; middle subplot), and their interaction (right subplot). (**B**) ROIs for the PPI analysis. (**C**) Results of PPI analysis for subject 1 (example). The slopes of the three straight lines represent the PPI values between dACC and left IOG in the three tasks. (**D**) PPI value between the dACC and each occipital ROI in three tasks (group effects). (FDR, two-sample *t*-test, *p* < 0.05).

**Figure 8 brainsci-13-01387-f008:**
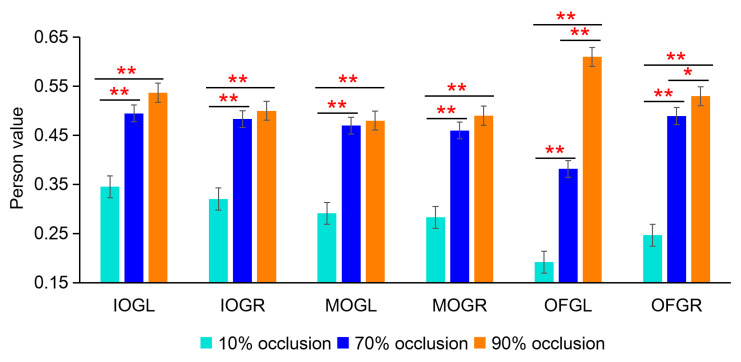
The first BOLD eigenvector analysis results. The Pearson correlation coefficients of the time series between the dACC and other brain regions at different occlusion levels (FDR, two-sample *t*-test: * = *p* < 0.05, ** = 0.01).

**Table 1 brainsci-13-01387-t001:** Variations in brain activation regions across diverse occlusion object recognition tasks.

Anatomical Regions and BA	Cluster Size (Voxels)	*t*-Score (FWE, *p* < 0.05)	x	y	z
90% occlusion > 10% occlusion					
L Inferior Occipital Gyrus (BA18)	67	6.3	−30	−84	−6
R Inferior Occipital Gyrus (BA18)	97	7.21	39	−84	−6
L Middle Occipital Gyrus (BA19)	47	5.71	−27	−87	12
R Middle Occipital Gyrus (BA19)	149	7.13	45	−78	0
L Dorsal anterior cingulate cortex (BA32)	42	5.9	−6	18	42
70% occlusion > 10% occlusion					
R Middle Occipital Gyrus (BA19)	149	7.13	−24	−87	−12
R Inferior Occipital Gyrus (BA18)	128	7.12	34	−78	−8
L Middle Occipital Gyrus (BA19)	222	8.75	−24	−89	9
R Middle Occipital Gyrus (BA19)	166	8.32	45	−75	0
L Occipital Fusiform gyrus (BA37)	50	6.12	−30	−69	−9
R Occipital Fusiform gyrus (BA37)	62	9.45	30	−81	−6
90% occlusion > 70% occlusion					
L Dorsal anterior cingulate cortex (BA32)	32	5.27	−30	48	12

Abbreviations: BA = Brodmann area.

## Data Availability

The data that support the findings of this study are available from the corresponding author upon reasonable request.

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
