# Peer review of "Brain Functional Representation of Highly Occluded Object Recognition"

_brainsci, 2023, doi:10.3390/brainsci13101387_

Round 1

Reviewer 1 Report

The manuscript entitled "Brain functional representation of highly occluded object recognition" presents an intriguing investigation into the neural mechanisms underlying the recognition of highly occluded objects, focusing on the role and interplay between the occipital lobe and the dACC. The methodologies employed, including the usage of fMRI studies and various analyses like GLM, MVPA, and PPI, are commendable. However, there are several areas where the manuscript could benefit from further clarification and elaboration.

Specific Comments:

1. Figure 5c and Statistical Corrections

It is unclear from the details provided whether the p-values presented in Figure 5c have been corrected for multiple comparisons. Given that the study involves multiple comparisons, it would be prudent to apply appropriate corrections to prevent type I errors. It is recommended to clearly mention the procedure followed for correcting multiple comparisons and to reassess the results with these corrections applied.

2. Significance Estimation between Conditions in MVPA Analysis

The methodology behind the estimation of significance between the occipital and (dACC & Occipital) conditions needs further clarification. It would be beneficial to adopt a statistically robust approach by assuming a null hypothesis where no significant difference exists between the conditions. Creating a null distribution by randomly sampling data from both conditions and comparing decoding accuracy against this distribution would provide a more sound basis for determining significance. Please revise this section to include a detailed explanation of the methods used to estimate the null distribution and the statistical tests employed.

3. PPI Analysis and Explained Variance

The PPI analysis section would benefit from a detailed exposition of the variance explained by the first (highest) BOLD eigenvector. Providing this data would offer deeper insights into the strength and reliability of the observed connections.

4. Controls Implemented in Decoding Analysis (MVPA)

The concerns regarding the controls implemented in the decoding analysis (MVPA), as highlighted in the expert comments, are valid. A more rigorous approach to the analysis would involve sampling an equal number of features for both conditions to compare decoding accuracy. This would ensure that the observed differences are genuinely due to the variables under study and not merely a result of an increased number of features in one condition. Please reconsider the analysis with a balanced number of features and report the decoding accuracies and FDR-corrected p-values.

Conclusion

Overall, the study holds significant promise in shedding light on the neural mechanisms underlying the recognition of highly occluded objects. By addressing the aforementioned concerns and suggestions, the manuscript can achieve a level of rigor that matches its potential impact in the field. Looking forward to seeing the refined version of this research.

Thank you for considering these suggestions, and I hope they prove helpful in enhancing the quality and impact of your study.

Please correct the minor spelling and grammatical errors present in the manuscript. 

Reviewer 2 Report

In the manuscript submitted for review, the Authors described the functional representation of the brain when recognizing objects with a high degree of occlusion. I found the subject of the manuscript interesting and the whole work is well thought out.  The Authors put a lot of work into the preparation of this extremely interesting work. The reader's attention is undoubtedly attracted by carefully prepared photos, thanks to which the work is clear with such a large amount of data.

My comments:

1. the purpose of the review of the experience is clearly defined, while the application does not answer the question of whether the assumptions resulting from the purpose of the study were confirmed; the conclusion is "too general";

2. There is no information on who authored the brain images/pictures

Reviewer 3 Report

The study explores the neural mechanisms underlying the recognition of highly occluded objects, shedding light on how the brain processes incomplete visual information. It underscores the interplay between visual processing and cognitive control regions in the brain and emphasizes the importance of functional connectivity in effectively recognizing objects under challenging visual conditions. In summary, this study contributes to our understanding of how the brain handles challenging visual tasks involving highly occluded objects.

While the study you described provides valuable insights into the neural mechanisms of recognizing highly occluded objects, it may have some drawbacks and limitations:

Sample Size: The study involved 66 participants, which is a reasonable sample size for many neuroimaging studies, but it may still be limited in terms of generalizability. Larger and more diverse samples could strengthen the findings.

Task-Specific Findings: The study focused on object recognition tasks with varying degrees of occlusion. While this is informative for understanding the neural mechanisms behind this specific task, it may not generalize to other cognitive processes or real-world situations.

Artificial Environment: fMRI studies are typically conducted in controlled and artificial environments. Participants may behave differently or use different cognitive processes when recognizing objects in real-world settings, which could limit the ecological validity of the findings.

Correlation vs. Causation: While the study identified correlations between brain regions and occlusion degree, it doesn't necessarily establish causation. It's possible that the observed brain activations are associated with object recognition but not the direct cause.

Temporal Resolution: fMRI has limited temporal resolution, meaning it can't capture rapid brain processes. The study may not reveal the precise timing of neural activations during object recognition, which could be essential for understanding the underlying mechanisms fully.

In conclusion, while the study offers valuable insights, it's essential to consider its limitations and the need for further research to corroborate and extend these findings.

Reviewer 4 Report

Firstly, I am writing to express my gratitude for the opportunity to review the research article “Brain functional representation of highly occluded object recognition”. I am honored to have been selected to contribute to the peer-review process for Brain Sciences.

I understand the critical importance of rigorous evaluation in academic research and am eager to lend my expertise to this process. I am confident that my analysis will be of value to the authors and help ensure that the work is of the highest quality.

Thank you for entrusting me with this important task. I look forward to the opportunity to provide a thorough and constructive review.

This study investigates the neural mechanisms involved in recognizing highly occluded objects by conducting functional magnetic resonance imaging (fMRI) experiments with 66 subjects. The results reveal that the activation levels of specific brain regions, including the occipital lobe and the dorsal anterior cingulate cortex (dACC), are related to the degree of object occlusion. Multivariate pattern analysis (MVPA) shows improved object recognition accuracy when considering dACC activation as a feature, highlighting the collaborative role of dACC and the occipital lobe in recognizing occluded objects. Additionally, psychophysiological interaction (PPI) analysis indicates enhanced functional connectivity between the dACC and the occipital lobe as occlusion increases, underscoring the importance of this connectivity in identifying highly occluded objects. These findings enhance our understanding of how the brain processes incomplete visual information during object recognition.

I would like to make a series of improvement suggestions to the authors:

INTRODUCCIÓN

Clarify the Specific Challenges of Highly Occluded Object Recognition: Provide a more detailed explanation of the specific challenges and complexities involved in recognizing highly occluded objects, highlighting the limitations of current computer vision algorithms and the need for a deeper understanding of human performance.

Discuss the Relevance of Human-Level Performance: Elaborate on why achieving human-level performance in highly occluded object recognition tasks is a significant benchmark, emphasizing the unique cognitive abilities of humans in this context.

Explore Existing Models and Theories: Discuss existing models and theories related to object recognition under occlusion in greater depth, including their strengths and limitations. This can help frame the research within the context of prior work.

Highlight the Role of Prefrontal Cortex (PFC): Emphasize the potential role of the prefrontal cortex (PFC) in highly occluded object recognition more explicitly, given its likely involvement in complex cognitive processes. Discuss relevant studies and findings that support this hypothesis.

Specify Research Objectives Clearly: Clearly outline the specific research objectives or questions that the study aims to address. This will help readers understand the focus and scope of the research more precisely.

Mention Novelty and Contribution: Explicitly state the novelty and contribution of the current research in advancing our understanding of highly occluded object recognition. Explain how the study extends or builds upon previous work.

Justify the Use of fMRI and Analytical Techniques: Provide a rationale for choosing functional magnetic resonance imaging (fMRI) and the specific analytical techniques (GLM, MVPA, PPI) used in the study, emphasizing how they are well-suited to address the research questions.

Consider Ethical and Practical Implications: Briefly touch upon any ethical considerations or practical implications of the research, such as its potential applications in improving computer vision algorithms or aiding individuals with visual impairments.

Acknowledge Potential Limitations: Acknowledge potential limitations of the study upfront, such as sample size, generalizability, or the specific tasks used for occluded object recognition. Addressing these limitations can strengthen the research.

Provide a Roadmap: Offer a brief roadmap of the paper to give readers an overview of the subsequent sections and how they contribute to answering the research questions.

METHOD

Diversity of Participants: Consider diversifying the participant pool to include individuals from various age groups and cultural backgrounds. This can help assess the generalizability of findings beyond university students.

Screening Procedures: Provide more details about the screening procedures for psychiatric or neurological disorders to ensure a rigorous selection process. Mention any standardized assessment tools or criteria used.

Control for Medication: Specify the duration of refraining from medication usage before the experiment and whether this was monitored during the study to ensure consistency.

Stimulus Validation: Include a section on stimulus validation, describing how the adjustments for size and contrast were made and providing measures of their effectiveness in ensuring similarity.

Task Familiarization: Explain in more detail how the practice tasks on an external computer were designed to avoid early familiarization with the target, and why this was important for the study.

Randomization Strategy: Clarify the randomization strategy used for presenting the blocks of different occlusion levels to ensure that the sequence of tasks did not bias the results.

Spatial Resolution: Justify the choice of the spatial resolution (2x2x2 mm³) for functional images, particularly in the context of investigating fine-grained neural representations.

Motion Correction: Provide information on how motion correction was performed, including any specific algorithms or procedures used to address head movement.

Multiple Comparison Correction: Specify the method used for correcting for multiple comparisons when defining the threshold for statistical significance (e.g., family-wise error correction or false discovery rate).

Feature Selection Rationale: Discuss the rationale behind selecting the specific brain regions (bilateral occipital lobes and dACC) as templates for the MVPA and how they relate to the research questions.

PPI Analysis Justification: Provide a brief explanation of why the generalized form of the PPI (gPPI) method was chosen and its advantages in studying functional connectivity during block-design experiments.

ROI Selection Criteria: Clarify the criteria used for selecting the specific ROIs (bilateral occipital lobes and dACC) and the justification for these choices.

FC Value Interpretation: Discuss the interpretation of the FC values between the dACC and occipital lobes in the context of the study's hypotheses.

RESULTS

In Behavioral Data:

Include Error Analysis: While average recognition accuracy and response times are reported, consider including a more detailed error analysis, such as the types of errors made (e.g., false positives, false negatives) and their relationship to occlusion levels.

Statistical Analysis: Provide specific statistical tests used to assess the significance of differences in accuracy and reaction times across occlusion levels, and report effect sizes if applicable.

GLM Analysis Results:

Quantitative Comparisons: Offer quantitative comparisons of the activation levels in different brain regions across occlusion levels, including effect sizes or percent signal change, to support the reported observations.

Discussion of Frontal and Parietal Activation: Elaborate on the functional significance of activation in the frontal and parietal lobes during the occluded object recognition tasks. What cognitive processes might these regions be associated with in this context?

Multivariate Pattern Analysis Results:

Detailed Confusion Matrix: Provide a more detailed confusion matrix for the multivariate pattern analysis, showing the classification performance for each task separately, along with metrics such as precision, recall, and F1-score.

Discussion of Classification Patterns: Interpret the classification patterns and discuss what they reveal about the discriminative power of occipital lobes and the dACC in differentiating between occlusion levels.

Statistical Significance: Specify whether the observed increases in classification accuracy with the addition of dACC features are statistically significant and discuss their implications in more detail.

Functional Connectivity: gPPI Analysis Results:

Discussion of Connectivity Patterns: Interpret the patterns of functional connectivity between the dACC and occipital lobes. Explain how these connectivity changes might relate to the processing of highly occluded objects.

Statistical Testing: Provide details on the statistical tests used to determine the significance of the observed differences in FC values across occlusion levels.

Biological Relevance: Discuss the biological and cognitive significance of increased functional connectivity between the dACC and occipital lobes in the context of highly occluded object recognition.

DISCUSSION

The Activation of the dACC and Occipital Lobe under Different Occluded Conditions:

Clarify the S-Type Activation Theory: Provide a more detailed explanation of the S-type activation theory in neural cytology and how it applies specifically to the observed activation patterns in the dACC and occipital lobe. Explain the theoretical underpinnings and supporting evidence in greater depth.

Link to Cognitive Processes: Discuss the cognitive processes associated with heightened activation in the dACC under 90% occlusion. How does this relate to decision-making or other cognitive functions, and what are the implications for understanding occluded object recognition?

The FC Between the dACC and Occipital Lobe:

Functional Integration Interpretation: Elaborate on the functional significance of the increased FC between the dACC and occipital lobe. How does this enhanced connectivity contribute to occluded object recognition, and what does it reveal about information processing strategies in the brain?

Task Difficulty and Brain Integration: Discuss the relationship between task difficulty, brain integration, and FC, drawing on previous research on cognitive tasks. How does the observed increase in FC align with theories of brain function during challenging tasks?

The Role of the dACC in Decision-Making Processes under Highly Occlusion:

Decision-Making Mechanisms: Provide a more detailed explanation of how the dACC's role in decision-making processes aligns with the experimental paradigm. Discuss the neural mechanisms by which the dACC facilitates decision-making under uncertainty, particularly in the context of occlusion.

Interpretation of Reaction Time and Accuracy: Relate the longer reaction times and lower accuracy under highly occluded conditions to the proposed decision-making role of the dACC. How do these behavioral findings support the neural mechanisms discussed?

CONCLUSION

Implications for Object Recognition: Summarize the practical implications of the study's findings for object recognition in real-world scenarios, particularly in situations with incomplete visual information. How might these insights be applied in fields such as computer vision or human-computer interaction?

Limitations and Future Directions: Acknowledge any limitations of the study and suggest potential avenues for future research. Are there specific aspects of occluded object recognition that remain unexplored or questions that arise from the current findings?

Continue the excellent effort; your commitment and diligence are clearly reflected in the caliber of your research. I am confident that, with some refinements, this manuscript will be prepared for submission.

Reviewing your work has been a delightful experience, and I am assured that, with the proposed revisions, your paper will significantly enrich the field. I extend my best wishes for your ongoing research endeavors and eagerly anticipate your forthcoming publications.

I'd like to convey my heartfelt gratitude for the dedication and hard work you've put into your research. The article stands to benefit from substantial enhancements, and I strongly recommend that the authors undertake a comprehensive revision to elevate its quality before submitting it again.

Regards,

Round 2

Reviewer 1 Report

I have reviewed the updated version of the document and find no further comments to provide.

Please proofread the manuscript as there are still grammatical and spelling errors.
For example, Figure 8 has a spelling error in the Y-axis label.